EMBO
Molecular Medicine

# A DNA/HDAC dual-targeting drug CY190602 with significantly enhanced anticancer potency

Chuan Liu[1,2,†,‡], Hongyu Ding[1,†], Xiaoxi Li[1,†], Christian P Pallasch[3], Liya Hong[4], Dianwu Guo[4], Yi Chen[5], Difei Wang[6], Wei Wang[7], Yajie Wang[2,*,‡], Michael T Hemann[8,**] & Hai Jiang[1,***]

## Abstract

Genotoxic drugs constitute a major treatment modality for human cancers; however, cancer cells' intrinsic DNA repair capability often increases the threshold of lethality and renders these drugs ineffective. The emerging roles of HDACs in DNA repair provide new opportunities for improving traditional genotoxic drugs. Here, we report the development and characterization of CY190602, a novel bendamustine-derived drug with significantly enhanced anticancer potency. We show that CY190602's enhanced potency can be attributed to its newly gained ability to inhibit HDACs. Using this novel DNA/HDAC dual-targeting drug as a tool, we further explored HDAC's role in DNA repair. We found that HDAC activities are essential for the expression of several genes involved in DNA synthesis and repair, including TYMS, Tip60, CBP, EP300, and MSL1. Importantly, CY190602, the first-in-class example of such DNA/HDAC dual-targeting drugs, exhibited significantly enhanced anticancer activity *in vitro* and *in vivo*. These findings provide rationales for incorporating HDAC inhibitory moieties into genotoxic drugs, so as to overcome the repair capacity of cancer cells. Systematic development of similar DNA/HDAC dual-targeting drugs may represent a novel opportunity for improving cancer therapy.

**Keywords** DNA repair; dual-targeting anticancer drug; HDAC; nitrogen mustard

**Subject Categories** Cancer; Pharmacology & Drug Discovery

## Introduction

A significant challenge for treating cancer is to enhance the efficacy of existing drugs. Genotoxic drugs represent an important branch of chemotherapy, which kill cancer cells by attacking cellular DNA. Double-strand breaks (DSBs) resulting from such attacks are extremely toxic, and one irreparable DSB is sufficient to induce cell death (Jackson & Bartek, 2009). Despite such potent lethality, DNA damage caused by anticancer drugs can be mitigated by cellular DNA repair machinery, thus enabling some cancer cells to survive and ultimately cause treatment failure. Drugs that inhibit DNA repair, such as ATM and DNAPK inhibitors, have been intensely investigated as potential means to improve chemotherapy (Hickson *et al*, 2004; Helleday *et al*, 2008; Willmore *et al*, 2008; Jiang *et al*, 2009).

Nitrogen mustards are a major type of genotoxic anticancer drugs. They work by attacking the N7 position of guanines on opposing DNA strands, thereby causing DNA interstrand cross-linkings (ICLs), which impede DNA replication forks and ultimately cause DSBs. Several pathways exist in mammalian cells to repair such damage, including the Fanconi anemia (FA) pathway, translesional synthesis (TLS), and homologous recombination (HR) (Knipscheer *et al*, 2009; Moldovan & D'Andrea, 2009). As a result, nitrogen mustards are generally well tolerated by cancer cells. Consequently, these drugs exhibit poor potency and limited treatment success. An emerging approach is to suppress DNA repair in order to enhance the efficacy of nitrogen mustards. Several lines of evidence support the potential benefits of this approach. For example, human patients with inherited deficiencies in FA pathway are extremely sensitive to ICLs (Taniguchi & D'Andrea, 2006), and tumors with mutations in the FA and HR pathways are hypersensitive to ICL agents (Van der Heijden *et al*, 2005; Kennedy &

1    Key Laboratory of Systems Biology, State Key Laboratory of Cell Biology, Institute of Biochemistry and Cell Biology, Shanghai Institutes for Biological Sciences, Chinese Academy of Sciences, Shanghai, China
2    Department of Oncology, Changhai Hospital, Second Military Medical University, Shanghai, China
3    Clinic for Internal Medicine, University Hospital of Cologne, Cologne, Germany
4    Hangzhou Minsheng Pharma Research Institute Ltd, Hangzhou, China
5    Crystal Biopharmaceutical LLC, Pleasanton, CA, USA
6    Department of Biochemistry and Molecular & Cellular Biology, Georgetown University Medical Center, Washington, DC, USA
7    Department of Chemistry, University of New Mexico, Albuquerque, NM, USA
8    The Koch Institute for Integrative Cancer Research at MIT, Massachusetts Institute of Technology, Cambridge, MA, USA
    *Corresponding author. Tel/Fax: +86 21 65383443; E-mail: yajiewa0459@163.com
    **Corresponding author. Tel: +1 617 253 3677; Fax: +1 617 253 8699; E-mail: hemann@mit.edu
    ***Corresponding author. Tel/Fax: +86 21 54921190; E-mail: hai@sibcb.ac.cn
    †These authors contributed equally to this work
    ‡Present address: Cancer Center, Shanghai East Hospital, Tongji University School of Medicine, Pudong, Shanghai, China

D'Andrea, 2006; Byrski *et al*, 2010). Efforts to screen for inhibitors of these repair pathways have yielded interesting results that may benefit future anticancer therapy (Chirnomas *et al*, 2006).

Bendamustine, a nitrogen mustard originally synthesized in the 1960s, recently regained great clinical interest due to its beneficial outcome in treating cancers (Keating *et al*, 2008; Cheson & Rummel, 2009; Knauf, 2009; Knauf *et al*, 2009; Garnock-Jones, 2010; Flinn *et al*, 2014). The recently approved clinical indications for bendamustine include chronic lymphocytic leukemia (CLL), small lymphocytic lymphoma (SLL), follicular lymphoma (FL), and mantle-cell lymphoma (MCL). However, the efficacy of bendamustine is limited by its poor drug potency. Therefore, bendamustine represents an interesting candidate for drug optimization. Here, we report our effort in developing and characterizing a more potent bendamustine derivative and argue for a knowledge-based redesign of existing genotoxic drugs.

## Results

To improve bendamustine's anticancer activity, a series of chemical derivatives were synthesized. Among these, CY190602 (Fig 1A, and referred to as CY thereafter) displayed 50- to 100-fold enhanced anticancer toxicity. Treatment of the NCI60 cell lines (http://dtp.nci.nih.gov/branches/btb/ivclsp.html) indicated a median growth inhibitor concentration ($GI_{50}$) of 2.2 μM for CY, in contrast to the median $GI_{50}$ of 77 μM for bendamustine (Fig 1B). Of note, the $GI_{50}$ for CY in MEFs was 57 μM, about 20-fold higher than its average $GI_{50}$ (3.2 μM) in NCI60 cell lines, suggesting that it may preferentially kill cancer cells. Comparing the $GI_{50}$ data of CY and bendamustine, such a significant increase in drug potency is rather surprising, given that CY differs from bendamustine only on its side chain, furthest away from the purported nitrogen mustard functional group. To understand CY's mechanism of action, we utilized a functional genetic approach for drug characterization (Jiang *et al*, 2011). Using this approach, we previously reported that despite the dramatic increase in potency, CY's primary mode of cell death induction is still nitrogen mustard-mediated DNA damage (Jiang *et al*, 2011).

To further understand how CY's chemical modifications might have resulted in such an increased potency, we synthesized compound A (Fig 1A), in which the chloride atoms of CY's nitrogen mustard group were substituted with hydroxyl groups. As a result, this compound retains the side chain modification of CY but lacks ability to attack DNA. Study of this compound would therefore enable us to focus on the side chain of CY. Consistent with the notion that CY primarily kills cancer cells through its nitrogen mustard group, compound A is very ineffective in killing cancer cells, even at 200 μM concentration (Fig 1C). Interestingly, despite its inability to kill cancer cells, compound A significantly synergized with bendamustine to kill cancer cells (Fig 1C). This suggests that compound A, and therefore the side chain of CY, is capable of enhancing the activity of bendamustine. Moreover, although compound A alone did not cause DNA damage, it significantly increased the level of γ-H2AX in bendamustine-treated cells (Fig 1D). Taken together, these results suggest that although the side chain group of CY is ineffective in killing cancer cells by itself, it is capable of enhancing the action of CY's nitrogen mustard group, which may explain CY's significantly increased anticancer potency.

Reexamination of CY's side chain suggested that it conforms to several rules of hydroxamic acid-based HDAC inhibitors (Suzuki *et al*, 2005). First, aromatic rings of CY facilitate its interaction with the surface of HDAC's enzyme pocket. Second, CY's 7-carbon side chain mimics the length of a lysine residue, which is optimal for inserting into HDAC's enzyme pocket. Third, CY's terminal hydroxamic acid group serves to chelate the zinc atom inside HDAC's enzyme pocket, thereby inhibiting HDAC's enzymatic activity. Indeed, when tested in cell-free HDAC activity assays, CY exhibited an $IC_{50}$ of 10–100 nM against HDAC1-6 *in vitro* (Fig 2A). In intact cells, CY and Cpd A also induced significant accumulation of acetylated histone H3 (Fig 2B and Supplementary Fig S1). In contrast, bendamustine lacks both the proper side chain length and the zinc-chelating hydroxamic acid group, which explains its inability to inhibit HDAC (Fig 2A and 2B, Supplementary Fig S1). Lastly, molecular docking of CY with the active sites of HDAC1 and HDAC2's X-ray structure suggested that CY is indeed capable of fitting into HDAC active sites (Fig 2C).

Taken together, these results suggest that CY is a dual-targeting drug, with both the DNA-damaging nitrogen mustard group and an HDAC inhibitory moiety. To address whether CY's HDAC inhibitory activity is essential for its anticancer activity, we synthesized compound B (Fig 1A), in which CY's hydroxamic acid group was substituted with a carboxylic acid group. Since carboxylic acid is far less effective in chelating zinc, this compound harbors little HDAC inhibitory activity (Fig 2D). We found that this compound is 10- to 20-fold less potent than CY (Fig 2E). This suggests that HDAC inhibitory group is essential for CY's enhanced anticancer activity. Given that both compound A (HDAC inhibitor only) and compound B (nitrogen mustard only) are far inferior in their ability to kill cancer cells, these results also suggested that these two functional groups synergize with each other to confer significantly enhanced anticancer potency. Such a synergy between nitrogen mustard and HDAC inhibitors was confirmed by treating cells with bendamustine and nontoxic dose of HDAC inhibitor Cpd A (Fig 2F and Supplementary Fig S2). To our knowledge, CY represents the first-in-class example of such DNA/HDAC dual-targeting drug that utilizes such intramolecular synergy.

Given that nontoxic doses of the HDAC inhibitor compound A were capable of enhancing DNA damage caused by bendamustine (Fig 2E), we hypothesized that the function of CY's nitrogen mustard group may be greatly potentiated by its HDAC inhibitory group. Next, we performed several experiments to address the mechanism of such potentiation.

Several recent publications argue for HDAC's crucial involvement in DNA repair (Miller *et al*, 2010; Robert *et al*, 2011). Although several mechanisms have been proposed, the full range of HDAC's function in DNA repair remains undefined. Given HDAC's important roles in regulating gene expression, we asked whether expressions of certain DNA repair genes are deregulated in CY-treated cells. According to current model, there are several core groups of ICL repair genes, including those participating in the ATR-Chk1, FA, TLS, and HR pathways (Knipscheer *et al*, 2009). In addition, since TLS and HR require active DNA synthesis, genes involved in dNTP production may also significantly impact ICL repair. To address how these groups of genes might be affected by HDAC inhibition, we treated cells with bendamustine, CY or HDAC inhibitor SAHA, and compared the expression level of more than 50

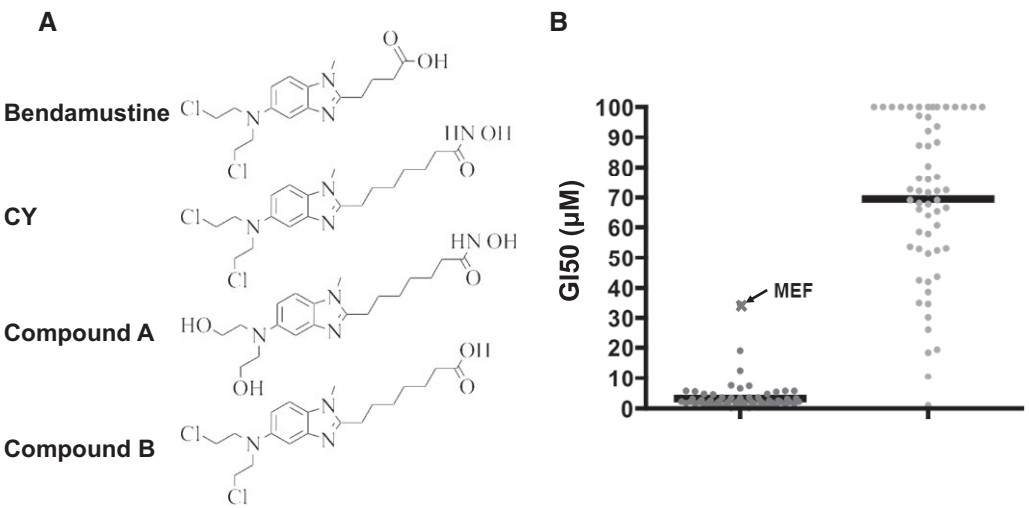

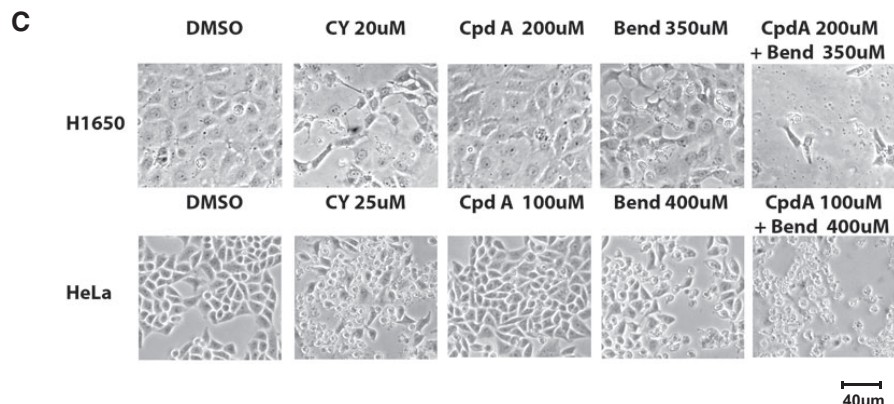

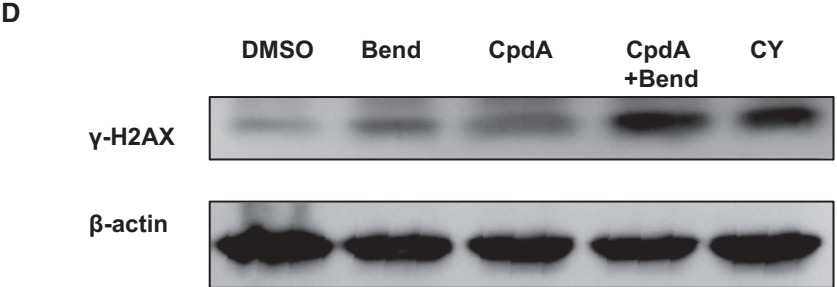

**Figure 1.  The side chain structure of CY enhances nitrogen mustard activity in cancer cells.**

A    Chemical structures of bendamustine, CY, compound A, and compound B (referred to as Cpd A and Cpd B hereafter).

B    Growth inhibitory concentration ($GI_{50}$) of CY and bendamustine in the NCI60 cell line panel.

C    Nontoxic dose of Cpd A enhances bendamustine cytotoxicity in cancer cells. Images were taken at 48 h posttreatment.

D    Cpd A augments DNA damage caused by bendamustine. H1650 cells were treated with 100 μM Bend, 50 μM Cpd A, 100 μM Bend plus 50 μM Cpd A, or 10 μM CY for 12 h, and cell lysates were blotted for γ-H2AX. Actin was used as a loading control.

Source data are available online for this figure.

genes belonging to these aforementioned gene groups (Supplementary Tables S1 and S2) as well as many other genes that have been reported to regulate DNA repair. Among these genes, we found that the expression levels of three histone acetyltransferases Tip60, CBP, and MORF, and MSL1, a gene associated with the histone acetyltransferases MOF (Smith *et al*, 2005; Huang *et al*, 2012), were all greatly suppressed in CY-treated cells as early as 6 h (Fig 3A), and such downregulation was not caused by drug-induced cell death

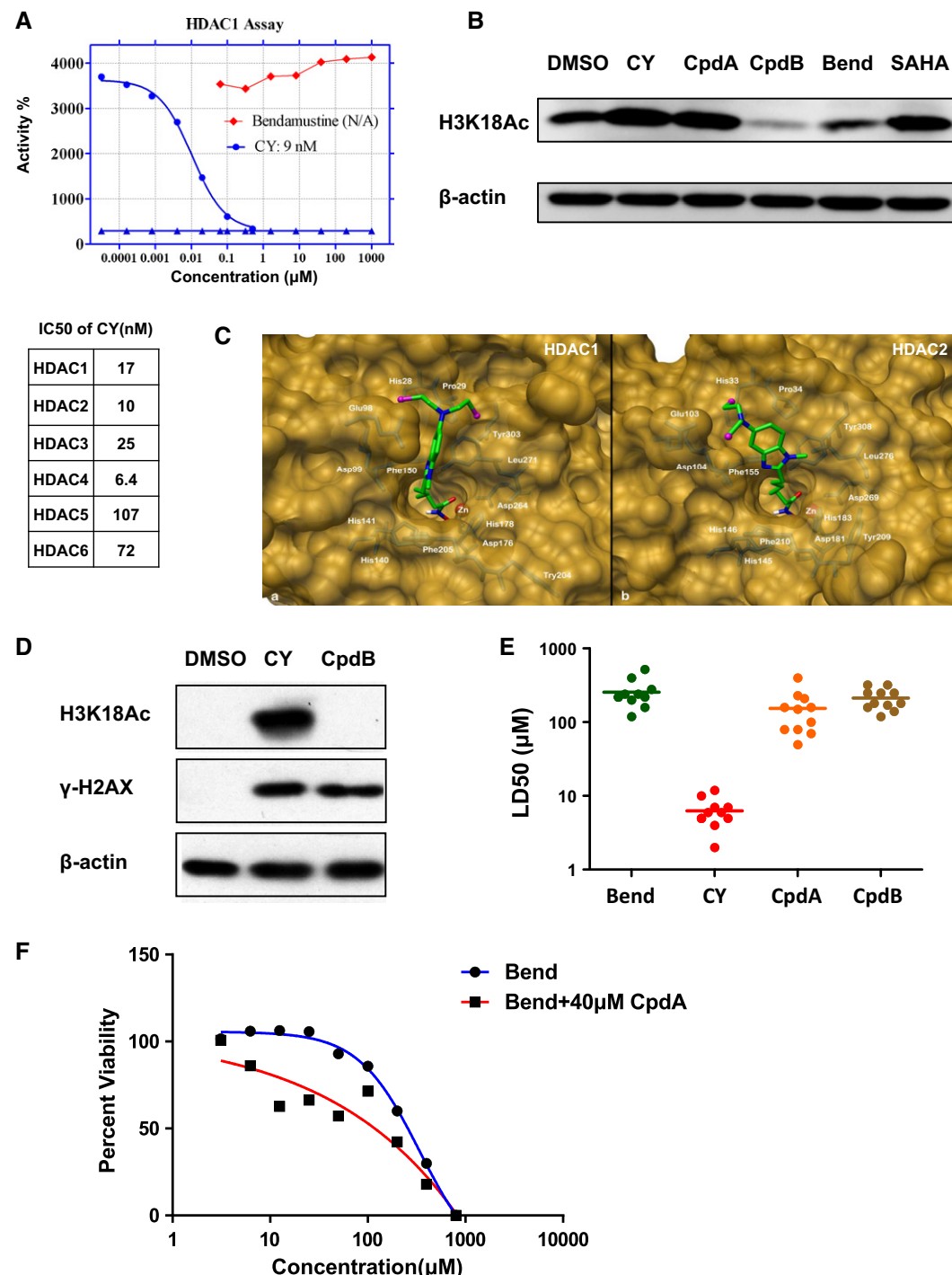

**Figure 2. CY harbors HDAC inhibitory activity that enhances nitrogen mustard's anticancer efficacy.**

A Upper panel, *in vitro* HDAC1 inhibition assay using bendamustine and CY. Lower panel, summary of CY $IC_{50}$ against other HDACs.

B CY and CpdA, but not bendamustine or CpdB, inhibited HDAC activities in cancer cells. H1650 cells were treated with 20 μM CY, 20 μM CpdA, 200 μM CpdB, 200 μM Bend, or 10 μM SAHA for 12 h, and cell lysates were blotted for histone acetylation markers. Actin was used as a loading control.

C Molecular docking of CY with the structure of HDAC1/2 enzyme pocket.

D Alteration of CY side chain (Cpd B) caused loss of HDAC inhibitory activity and resulted in reduced ability to induce DNA damage. Cells were treated for 12 h. Cells treated with 200 μM Cpd B and 20 μM CY exhibited similar levels of γ-H2AX. Actin was used as a loading control.

E Summary of lethal dose 50 ($LD_{50}$) of bendamustine, CY, Cpd A (HDAC inhibition only), and Cpd B (nitrogen mustard only) against 11 human cancer cell lines.

F Cpd A enhances the anticancer activity of bendamustine. H1650 cells were treated with various doses of bendamustine with or without 40 μM Cpd A (nontoxic dose, see Fig 1C and Supplementary Fig S2) for 48 h and subjected to MTT viability assays.

Source data are available online for this figure.

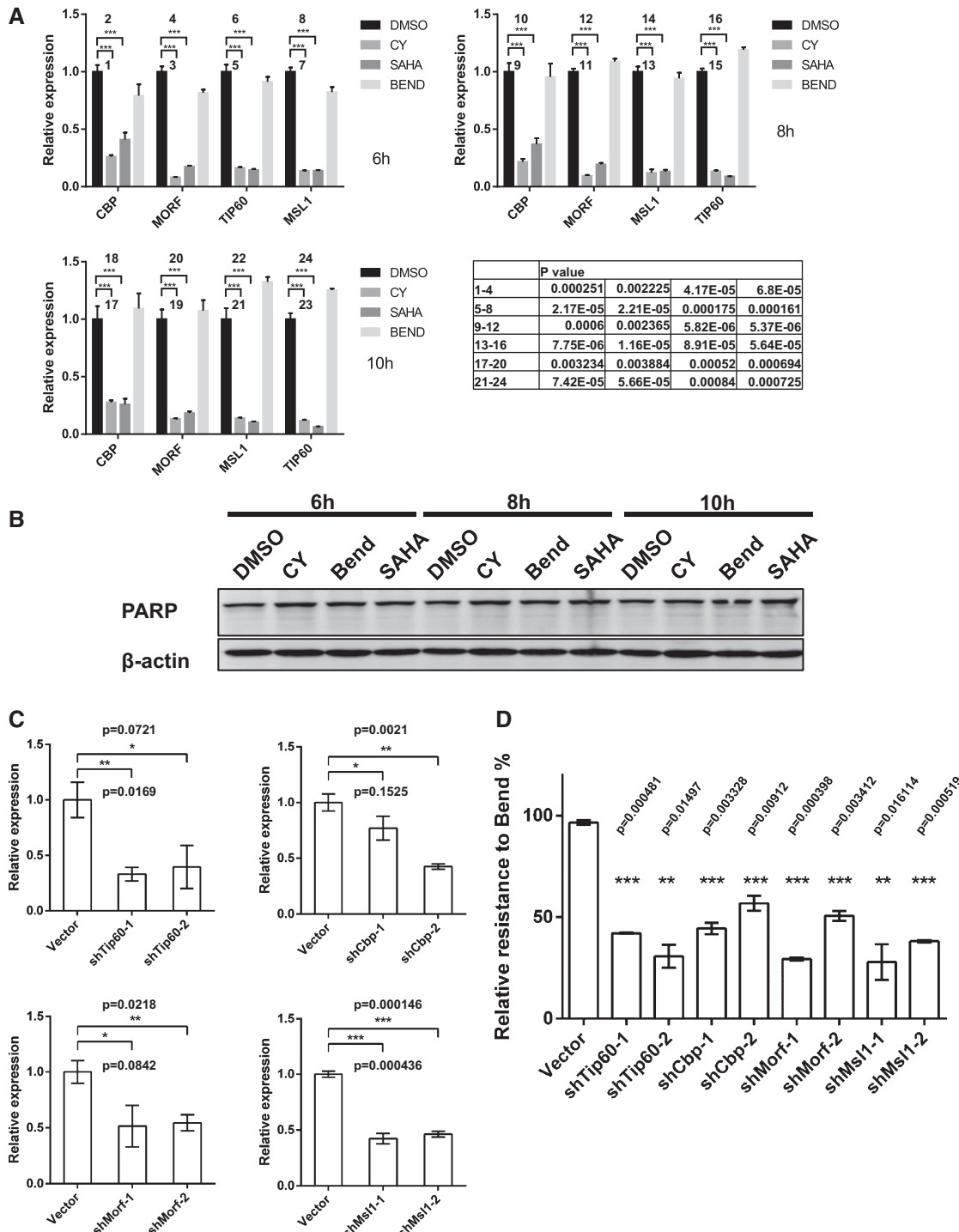

**Figure 3.  HDAC inhibition by CY leads to downregulation of several histone acetyltransferases involved in DNA repair.**

A   CY, but not bendamustine treatment, led to suppression of CBP, TIP60, MORF, and MSL1 in H1650 cells. Cells were treated with DMSO, bendamustine (350 μM), CY (15 μM), or SAHA (10 μM) for 6 h, and mRNA was extracted for qPCR analysis. Data represent mean ± SEM from three independent experiments, and statistical significance was determined by unpaired two-tailed $t$-test. ***$P < 0.01$.

B   Drug treatment up to 10 h at the concentrations indicated in (A) did not cause apoptosis as judged by PARP cleavage. Actin was used as a loading control.

C   ShRNA-mediated suppression of CBP, TIP60, MORF, and MSL1 at mRNA level. Data represent mean ± SEM from three independent experiments, and statistical significance was determined by unpaired two-tailed $t$-test. *$P < 0.1$, **$P < 0.05$, ***$P < 0.01$.

D   Suppression of CBP, TIP60, MORF, and MSL1 sensitized cells to bendamustine. $Y$-axis represents relative resistance calculated from results of GFP competition assays, and statistical significance was determined by unpaired two-tailed $t$-test. Data represent mean ± SEM from two independent experiments. **$P < 0.05$, ***$P < 0.01$.

Source data are available online for this figure.

   

(Fig 3B). Tip60 has been reported to regulate ATM-mediated DNA repair (Sun *et al*, 2005; Kaidi & Jackson, 2013), whereas CBP have been shown to regulate ATR-Chk1 pathway and chromatin remodeling at DNA lesions (Hasan *et al*, 2001; Stauffer *et al*, 2007). MSL1 has also been shown to affect DNA repair (Gironella *et al*, 2009; Aguado-Llera *et al*, 2013). Importantly, shRNA suppression of these four genes (Fig 3C, Supplementary Fig S3) each sensitized cells to bendamustine (Fig 3D, Supplementary Tables S3 and S4). Of note, suppression of these genes alone did not affect cellular viability without drug treatment (Supplementary Fig S4). Taken together, our data demonstrated that in addition to other reported mechanisms (Miller *et al*, 2010; Robert *et al*, 2011), HDAC inhibition could increase the potency of nitrogen mustards through downregulation of genes that regulate DNA repair, including Tip60, CBP, MORF, and MSL1. Our study of CY as a prototype of DNA/HDAC dual-targeting drug demonstrates that by incorporating HDAC inhibitory moiety into traditional DNA-damaging drugs, it is indeed possible to achieve much higher toxicity against cancer cells.

Lastly, we studied the antitumor activity of this dual-targeting drug *in vivo*. We first determined that the MTD (maximally tolerated doses) of CY is 60 mg/kg in mice (Fig 4A). Next, we used a transplantable BCR-ABL-driven acute lymphoblastic leukemia (ALL) mouse model to assess CY's *in vivo* activity. BCR-ABL-positive ALL accounts for about 1/3 of adult human ALL cases and is traditionally treated with many types of chemotherapeutics. Despite the use of heavy chemotherapy regimen, patients with this disease have a very poor survival rate (Stock, 2010). Treatment with targeted therapeutics that inhibit BCR-ABL, such as dasatinib, is an emerging therapy approach for this disease (Yanada *et al*, 2009). We chose this mouse model therefore in order to compare CY with a wide range of traditional chemotherapeutics as well as targeted therapeutics in an *in vivo* setting.

In cultured BCR-ABL ALL cells, CY is more active than bendamustine (Fig 4B). Mice transplanted with BCR-ABL cells died around 12 days after the injection of 40,000 leukemia cells without treatment (Fig 4C). Treatment with SAHA or bendamustine at MTD extended survival by approximately 2 and 5 days, respectively. In contrast, mice treated with CY showed average survival extension of 14 days (Fig 4C). Moreover, when compared with other chemotherapeutic agents commonly used in BCR/ABL ALL human patients, including cyclophosphamide, doxorubicin, cytarabine, and vincristine, CY's survival extension was also superior to these drugs (Fig 4D). This indicated that the DNA/HDAC dual-targeting approach confers better therapeutic outcome and may have possible clinical advantages.

In both mouse models and human patients, BCR-ABL ALL can be effectively managed by continuous administration of the BCR-ABL/Src inhibitor dasatinib (Gruber *et al*, 2009; Boulos *et al*, 2011). In addition, a recent report indicated potential efficacy of mTOR inhibitor PP242 in BCR-ABL ALL mouse model (Janes *et al*, 2010). Next, we compared CY's efficacy with targeted therapeutics dasatinib and PP242 in this ALL model. To access the long-term clinical benefit of CY treatment, we developed a weekly CY treatment schedule that was well tolerated. Consistent with existing report (Janes *et al*, 2010), daily treatment of the mTOR inhibitor PP242 extended the mice survival by an average of 8 days (Fig 4E). In contrast, daily doses of dasatinib extended the survival up to 2 months. Importantly, weekly doses of CY prolonged the survival to a similar extent as dasatinib (Fig 4E), further demonstrating CY's potent anticancer effects *in vivo*. Lastly, in xenograft models using the human lung cancer cell line H460, CY also showed potent anticancer activity that was superior to bendamustine (Fig 4F and G). Taken together, these data showed that the DNA/HDAC dual-targeting drug CY has potent anticancer activity *in vivo*.

Given that bendamustine has shown significant clinical efficacies in treating chronic lymphocytic leukemia (CLL), we further tested CY's efficacy in freshly isolated human CLL cells. CY killed nearly all CLL cells at 5 μM. In contrast, bendamustine, and fludarabine, another CLL frontline drug, at 100 μM could only kill about 80% CLL cells (Fig 5A). Moreover, despite CY's high efficacy in killing CLL cells at 5 μM, healthy primary B cells still remain about 40% viable after treatment with 20 μM CY (Fig 5B). These data show that CY potently kills CLL cells, and there is a preferential killing of transformed cells over healthy, non-transformed B cells.

In clinics, bendamustine is ineffective in treating CLL cells with 17p deletion. CLL cells harboring this deletion lose p53 and become refractory to bendamustine treatment (Zaja *et al*, 2013). Interestingly, CY kills 17p-deleted and 17p-retaining CLL cells with similar efficacy (Fig 5C), suggesting that CY may represent a potential new choice for exploring treatment strategies for 17p-deleted CLLs.

## Discussion

In this report, we described the development and characterization of CY, a bendamustine-derived, DNA/HDAC dual-targeting anticancer drug. In our previous report (Jiang *et al*, 2011), we tested CY in several cell lines and found it to exhibit significantly enhanced anticancer activity *in vitro*; however, the source of such enhanced activity remained undefined. One important question is whether

---

**Figure 4.  CY exhibited enhanced anticancer activity *in vivo*.**

A     Determination of maximally tolerated dose (MTD) of CY at 60 mg/kg. Other drugs were used at MTD according to the literature. For each drug, *n* = 5 and all mice survived treatment. Data represent mean ± SEM.

B     CY exhibited enhanced activity against BCR-ABL Arf$^{-/-}$ murine ALL cells *in vitro*.

C, D   CY showed superior antitumor activity compared with bendamustine, SAHA, and other chemotherapeutic drugs in mice transplanted with BCR-ABL ALL cells. *P*-values of CY versus other drugs: NT (*P* = 5.02E-06) (C), SAHA (*P* = 4.45E-06), Bend (*P* = 5.18E-06), NT (*P* = 1.20E-05) (D), VCR (*P* = 1.13E-05), Dox (*P* = 8.08E-06), AraC (*P* = 7.62E-06), and CTX (*P* = 1.13E-05). Survival statistical analysis was done with the Mantel–Cox (log-rank) test of GraphPad Prism.

E     Therapeutic effects of CY (weekly dose) and targeted drugs PP242 (daily dose) and dasatinib (daily dose) in mice transplanted with BCL-ABL ALL cells. *P*-values of CY versus other drugs: NT (*P* = 3.94E-05), PP242 (*P* = 1.20E-05), and dasatinib (*P* = 0.99). Survival statistical analysis was done with the Mantel–Cox (log-rank) test of GraphPad Prism.

F, G   Antitumor activity of CY and bendamustine in nude mice models. Mice were inoculated with human NSCLC cell line H460. When tumor volume reached 200 mm$^3$, mice were treated with single dose of bendamustine (40 mg/kg, MTD) or CY (20, 40, 60 mg/kg). In (G), tumors were dissected out after 15 days posttreatment.

---

   

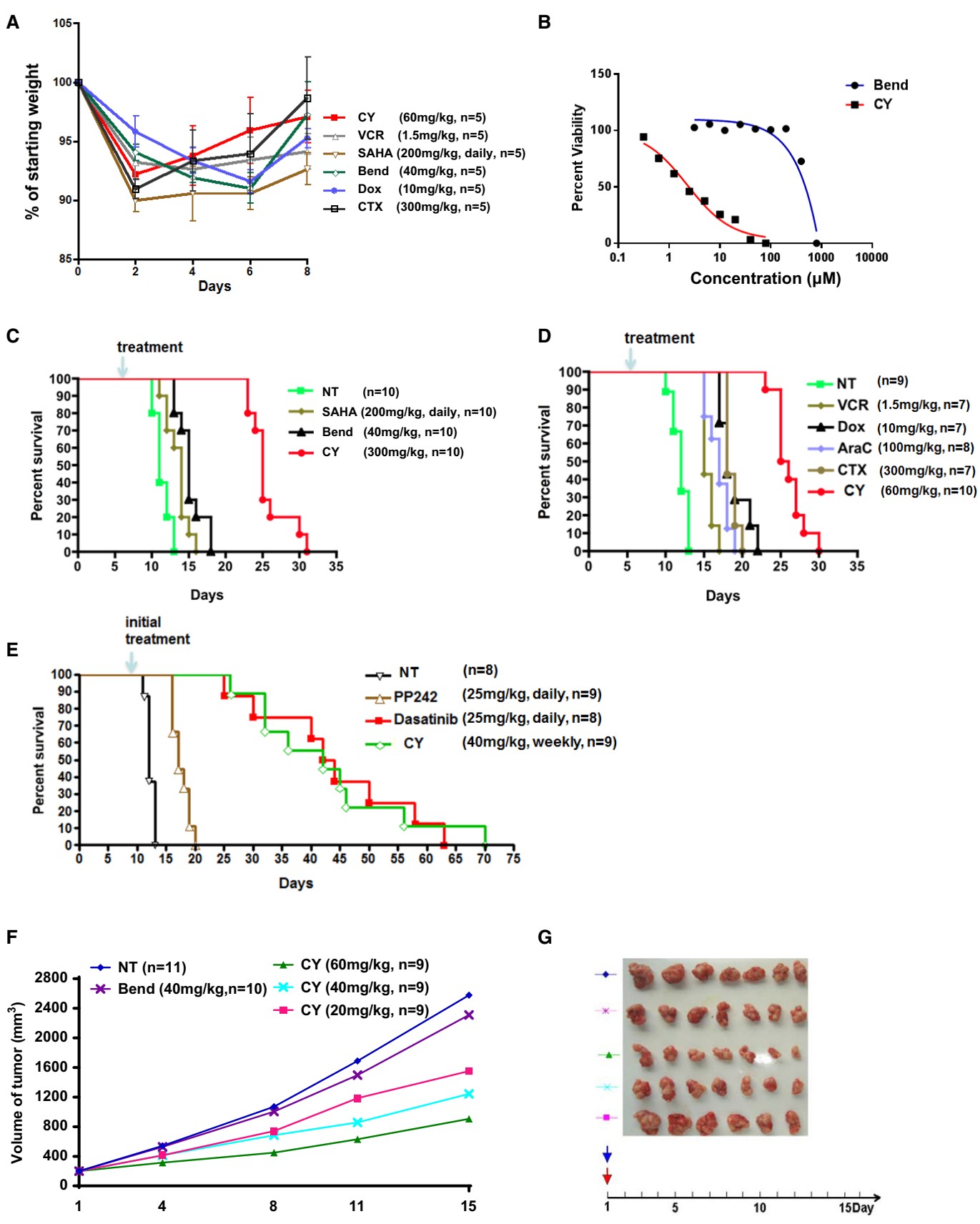

**Figure 4.**

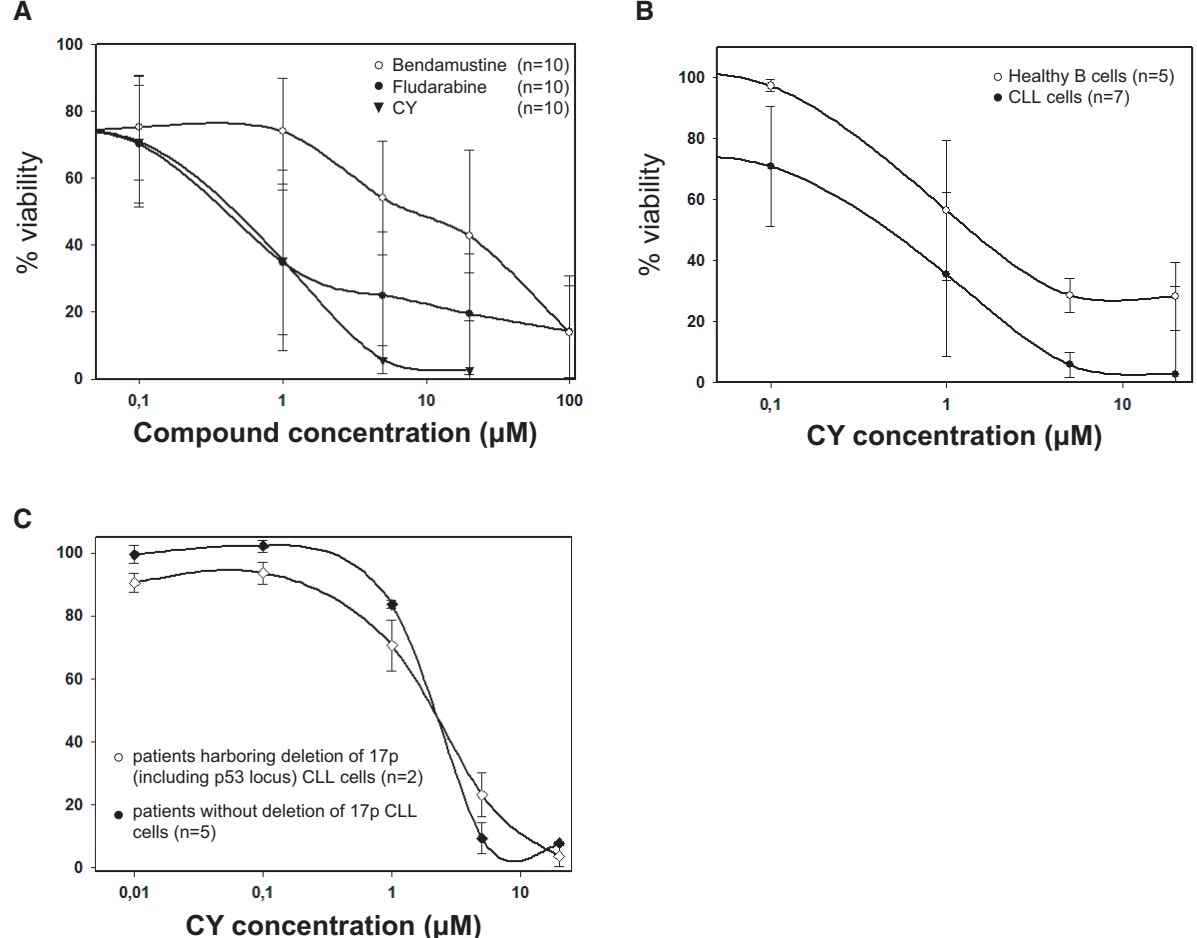

**Figure 5.  CY exhibited activity with CLL samples.**

A  Dose curve of CY, fludarabine, and bendamustine using fresh CLL samples. Data represent mean ± SD.
B  CY preferentially killed CLL cells over healthy B cells. Data represent mean ± SD.
C  Toxicity of CY190602 in patients harboring deletion of chromosome 17p (including p53 locus) compared to patients without del17p. Viability of samples was normalized to background apoptosis for individual patient samples. For all experiments, cell viability was determined after 72 h of drug treatment. Data represent mean ± SD.

there are generally applicable rules that we can derive from CY, so that we can apply such rules to improve other existing anticancer drugs. In this report, we dissected this phenomenon and provided mechanistic explanation. Biochemically, CY is capable of both inducing DNA damage and inhibiting HDACs (Figs 1D and 2B). In a panel of 60 commonly used cancer cell lines, CY exhibited 50- to 100-fold increased activity compared to the parental compound bendamustine. *In vivo*, CY also exhibited far-superior anticancer efficacy compared to bendamustine. Given that bendamustine has recently been shown to exert strong clinical activity in several types of human malignancies, it should be interesting to test CY's efficacy in these disease settings.

One interesting question remains whether CY would be superior to the combination treatment of bendamustine and HDAC inhibitors. Due to both activities being tethered in the same molecule of CY, it is possible that by delivering both DNA damaging and HDAC inhibitory ability at the same time to tumor cells *in vivo*, CY may work better than the combinatory treatment of bendamustine and HDAC inhibitor, whose pharmacodynamics and pharmacokinetics

differ *in vivo*. Unfortunately due to the difference in treatment schedule (bendamustine, single dose; SAHA, daily dose) and toxicity issues, we could not establish an effective treatment schedule that involves both bendamustine and SAHA. Considering that there is currently no protocol using both bendamustine and SAHA in clinics and that bendamustine is effective in several cancer types as a single drug, we therefore focused on comparing the efficacy between CY and bendamustine *in vivo*. Our results suggested that CY had significantly enhanced *in vivo* anticancer activity compared to bendamustine in several transplanted and xenograft cancer models. Moreover, the BCR-ABL mouse ALL model enabled us to compare the efficacy of CY with several other commonly used anticancer drugs. Our results showed that CY has superior anticancer activity over several first-line chemotherapeutic drugs, and in this model CY's efficacy is even comparable to the targeted drug, BCR-ABL inhibitor dasatinib. Taken together, the data suggested that this novel DNA/HDAC dual-targeting drug CY has significant anticancer efficacy *in vivo*.

Although HDAC inhibitors have been investigated as radio-sensitizing agents (Ree *et al*, 2010), the complete picture of HDAC

in DNA repair remained elusive. Several recent publications shed light on this topic and suggested that HDAC may function directly at sites of DNA damage by altering local histone code (Miller *et al*, 2010; Robert *et al*, 2011). In addition, a recent proteomic study found that many DNA repair proteins including MDC1, BLM, and Rad50 are modified by acetylation after HDAC inhibition (Choudhary *et al*, 2009). It is possible that HDAC inhibition may negatively impact the stability and/or activity of these repair proteins. For example, it was shown that upon HDAC inhibition, the HR nuclease Sae2/CtIP is acetylated and degraded (Robert *et al*, 2011). In addition, HDAC inhibition can suppress the ATM signaling pathway, thereby sensitizing cancer cells to DNA damage (Thurn *et al*, 2013).

In this report, we focused on HDAC's role in DNA repair by demonstrating its rapid transcriptional control of other groups of important DNA repair genes. We examined gene expression level after 6-h treatment with CY or SAHA in multiple types of cancer cell lines. Interestingly, among genes whose expression is significantly suppressed upon HDAC inhibition, Tip60, CBP, MORF, and MSL1 are all histone acetyltransferases or histone acetyltransferase-associated protein, and shRNA suppression of these genes sensitized cells to DNA damage. Moreover, we also found that upon HDAC inhibition, another histone acetyltransferase EP300, and TYMS, a gene involved in nucleotide synthesis, are both transcriptionally downregulated upon HDAC inhibition (Supplementary Fig S5), and shRNA suppression of either gene by itself is lethal to cancer cells even without DNA damage (Supplementary Fig S6). Because cells with TYMS or EP300 shRNAs were rapidly eliminated in cell culture, we could not analyze whether loss of these two genes further sensitized cells to DNA damage. However, given their important roles in nucleotide pool maintenance and DNA repair (Hasan *et al*, 2001), it is rather possible that downregulation of these two genes upon HDAC inhibition could cause further impairment in DNA repair.

Of note, previous reports have suggested that BRCA1 and the NHEJ repair genes Ku70, Ku80, and DNAPKcs are suppressed by HDAC inhibitors (Zhang *et al*, 2007, 2009). In our hand, unlike the case for Tip60, CBP, MORF, MSL1, and EP300, suppression of BRCA1, Ku70, Ku80, and DNAPKcs did not occur at 6 h upon HDAC inhibition, suggesting that it may not be an early response upon HDAC inhibitor treatment. To further examine this discrepancy, we searched the connectivity map (Lamb *et al*, 2006), a consortium of microarray data, and identified 12 microarray datasets from cells treated by HDAC inhibitors. Importantly, none of the previously reported genes (BRCA1, Ku70, Ku80, and DNAPKcs) were among the top 200 downregulated genes in any of these HDAC inhibitor-treated cells. In contrast, Tip60, CBP, MORF, MSL1, and EP300 were among the top 200 downregulated genes in 5, 6, 11, 9, and 5 of the 12 HDAC inhibitor-treated cells, suggesting that this is a highly potent and consistent effect of HDAC inhibitors. Importantly, given our finding that HDAC inhibition caused rapid and significant downregulation of these histone acetyltransferases or histone acetyltransferase-associated protein, it is possible that upon HDAC inhibition, a transcriptional feedback mechanism is activated to downregulate cellular acetyltransferase activity, which subsequently caused impairment of cellular DNA repair capacity. This may constitute a major transcription-related mechanism that contributes to HDAC inhibitor-mediated repression of DNA repair.

Given the potent lethality of irreparable DNA strand breaks, approaches to suppressing DNA repair have been intensively investigated, as they may bring significant benefits to cancer therapy. The recent success of PARP inhibitors in treating BRCA-deficient tumors showcases the therapeutic potential of such approach. Inhibitors of kinases with long-established roles in DNA repair, such as ATM, ATR, and DNAPKcs, are in various stages of preclinical and clinical investigations for their potential benefits in improving traditional chemotherapy. Moreover, with the recent advances in our understanding of DNA repair, several additional groups of enzymes have been recognized for their involvement of DNA repair, including HDACs (Miller *et al*, 2010; Robert *et al*, 2011), histone acetyltransferases (Sun *et al*, 2009; Niida *et al*, 2010), ubiquitin ligases (Kolas *et al*, 2007; Stewart *et al*, 2009), deubiquitinases (Nakada *et al*, 2010), SUMO ligases (Galanty *et al*, 2009; Morris *et al*, 2009), and histone methyltransferases (Liu *et al*, 2010). Pharmacological targeting of these enzymes may also enhance the efficacy of traditional genotoxic anticancer drugs. Our study of CY as a prototype of DNA/HDAC dual-targeting drug demonstrates that by incorporating small enzyme inhibitory moiety into traditional DNA-damaging drugs, we can achieve higher toxicity against cancer cells. Importantly, our result showed that it is structurally compatible to incorporate small enzyme inhibitory chemical moieties into DNA damage drugs, and such modifications can significantly enhance nitrogen mustard's anticancer activity. On the basis of this rationale, we have developed a novel nitrogen mustard derivative that targets both DNA and CDK, and it also showed about 100-fold increases in anticancer efficacy. This suggests that it may be generally applicable to incorporate various enzyme inhibitory moieties into traditional genotoxic drug to achieve better efficacy. Taken together, it is interesting to develop other types of HDAC/DNA dual-targeting drugs, as well as other types of drugs that target both DNA and some of the enzymes involved in DNA repair. Such drugs may by itself improve cancer treatment, and their much-improved anticancer efficacy also offers new possibilities for antibody-coupled, tumor-targeted drug delivery research. This may provide several novel categories of anticancer drugs for clinical investigation.

## Materials and Methods

### Cell culture and reagents

Cell lines were cultured using standard protocols provided by ATCC. Myc Arf$^{-/-}$ cells were cultured as described (Jiang *et al*, 2009). BCR-ABL mouse ALL cells were a kind gift from Dr. Richard Williams. CY190602 and its derivatives compound A and B were synthesized by Dr. Wang's Laboratory (U. New Mexico). Other chemicals were purchased from Selleck, EMD, or VWR. Antibodies against γ-H2AX (Millipore), actin (Sigma), H3K18Ac, H3K9Ac, and H3K56Ac (Cell Signaling) were used for Western blotting.

### shRNA construct cloning, RNA preparation and qPCR

Retroviral pMSCV-IRES-GFP vector and the cloning procedures have been previously described (Jiang *et al*, 2009, 2011). Target-gene knockdown efficiency was analyzed by qPCR. Total mRNA of cells was extracted with TRIzol reagent (Invitrogen, Carlsbad, CA, USA)

according to the manufacturer's instruction. Total mRNA was transcribed to cDNA with the SuperScriptIII First-Strand Synthesis System (Invitrogen, Carlsbad, CA, USA).

### Cell viability assays and determination of relative drug resistance

Cells were seeded in 96-well plates, treated with different concentrations of drugs for 48 h, and cell viability was analyzed by MTT assays according to manufacturer's protocols. Experiments to determine $GI_{50}$ with the NCI60 panel cell lines (http://dtp.nci.nih.gov/branches/btb/ivclsp.html) were performed at NCI (Alley *et al*, 1988; Shoemaker, 2006).

To test how shRNA suppression of certain genes affects cellular sensitivity to drugs, we used a GFP-based competition experimental system previously described by Jiang *et al* (Jiang *et al*, 2011). Briefly, shRNA and GFP were stably transduced into cells via retroviral vectors; therefore, cells in which targeted genes were knocked down also express GFP. A mixture of knockdown cells (GFP positive) and control cells (no viral infection, GFP negative) was treated with drugs. If knockdown of target gene sensitizes cells to drug, then in the survival cell population, the percentage of GFP-positive (gene knockdown) cells will decrease. By comparing GFP percentages with and without drug treatment, we can calculate relative resistance or sensitivity caused by target-gene knockdown, using a method previously described.

### Mouse experiments

Experimental procedures were approved by the Animal Care and Use Committee of Shanghai Institute of Biochemistry and Cell Biology, Chinese Academy of Sciences. In all, 200 wild-type C57BL/6 mice (6 weeks old, female) were used for determining maximally tolerated dose of CY, and testing CY and other drugs' efficacy in the BCR-ABL ALL model. BCR-ABL cells have been previously described (Williams *et al*, 2006). For *in vivo* experiments, 1 million cells were injected into mice via tail vein (Williams *et al*, 2006). At 7 days postinjection, mice were treated with indicated drugs at their MTDs. Mice were monitored daily for survival after drug treatment. Survival curve and statistical analysis were done using the Prism software. For xenograft H460 model, in all, nude mice (8 weeks old, female) were used to test CY and bendamustine's *in vivo* efficacy. The numbers of mice used in each experimental group are labeled on Fig 4.

### CLL patients and cells

This study was approved by the ethics committee of the University of Cologne (approval 01-163). Blood samples were obtained from patients fulfilling diagnostic criteria for CLL with informed consent according to the Helsinki protocol. Only patients without prior therapy or at least 12 months without prior chemotherapy were included in this study. Fresh blood samples were enriched by applying B-RosetteSep (StemCell Technologies, Vancouver, Canada) to aggregate unwanted cells with erythrocytes and Ficoll-Hypaque (Seromed, Berlin, Germany) density gradient purification resulting in purity > 98% of $CD19^+/CD5^+$ CLL cells. CLL cells were characterized for CD19, CD5, CD23, FMC7, CD38, ZAP-70, sIgM, sIgG, CD79a, and CD79b expression on a FACSCanto flow cytometer (BD

**The paper explained**

**Problem**

Most existing anticancer drugs attack single targets in cancer cells, which, however, activate various strategies to repair the damage induced by the drugs or to counter their efficacy. This significantly limits the efficacy of existing drugs and can cause treatment failure and illustrates the need for developing novel, more potent anticancer drugs.

**Results**

We developed and characterized a novel dual-targeting drug based on side chain modification of the nitrogen mustard drug bendamustine, with high potency against cancer. This drug not only attacks DNA, but also inhibits HDACs, a group of enzymes important for DNA repair, so that cancer cells cannot readily repair the damage it causes. We also show that this dual-targeting drug has significantly improved efficacy over drugs that have single targets in many cancer cell lines and various cancer mouse models.

**Impact**

Our results indicate that by incorporating two cooperating anticancer chemical groups into the same compound, dual-targeting drugs with higher efficacy can be generated. This approach can therefore be used to systematically increase the potency of traditional DNA-damaging drugs. Such dual-targeting drugs may provide new categories of anticancer drugs for cancer treatment.

PharMingen, Heidelberg, Germany). Controls were isolated from healthy blood donors using anti-CD19 MACS beads (Miltenyi, Bergisch Gladbach, Germany) resulting in at least 95% $CD19^+$ B cells.

### Apoptosis and cell viability assays

Apoptosis was determined by flow cytometry using Annexin V-FITC/7AAD staining (BD PharMingen) after 72 h. The cellular potency of compounds as defined by half-maximal induction of apoptosis in primary CLL cells was determined using concentrations up to 100 μM. The fraction of viable cells was determined by counting annexin V/7-AAD double-negative cells for each individual dosage. Median values were subsequently applied for regression analysis and calculation of the half-maximal dosage effect ($IC_{50}$). Curve fitting was performed using SigmaPlot (SPSS, Chicago). $IC_{50}$ values were determined by fitting data to the Hill equation $y = y0 + (axb)/(cb+xb)$.

**Supplementary information** for this article is available online: http://embomolmed.embopress.org

### Acknowledgements

We thank the Developmental Therapeutics Department of the NCI/NIH for testing CY190602 (NSC#751447) in their NCI60 anticancer screening program and for supplying bendamustine HCl salt. This work was supported by the major scientific research project (2013CB910404), the National Natural Science Foundation of China (31371418; 81372854; 81102010; 81202096), and Changhai Hospital 1255 discipline construction projects (No. CH12553 0400). The funders had no role in study design, data collection and analysis, decision to publish, or preparation of the manuscript. M.T.H. is the Eisen and Chang Career Development Associate Professor of Biology and is

supported by NIH RO1 CA128803 and the Ludwig Center for Molecular Oncology at MIT.

## Author contributions

HJ, MTH and YC were involved in study conception and design. HJ and MTH contributed to the development of methodology. CL, HD, XL, LH, CPP, DG, YC, DW and WW were involved in acquisition of data such as provision of animals and facilities, and acquisition and management of patients. HJ, CL, HD and CPP were involved in analysis and interpretation of data, such as statistical analysis, biostatistics, and computational analysis. HJ, MTH and YW participated in the writing, review, and/or revision of the manuscript. HJ, MTH and YW participated in the study supervision.

## Conflict of interest

Yi Chen is an employee of Crystal Biopharmaceutical LLC, a cancer drug discovery and development company located in Pleasanton, CA, USA. Dianwu Guo is the CSO of Hangzhou Minsheng Pharma Research Institute Ltd, who is developing CY190602 in China market. Liya Hong currently works in Zhejiang Institute for Food and Drug Control.

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
