## [Review Process File · EMBO Molecular Medicine]

A DNA/HDAC dual-targeting drug CY190602 with significantly enhanced anticancer potency

Chuan Liu, Hongyu Ding, Xiaoxi Li, Christian P. Pallasch, Liya Hong, Dianwu Guo, Yi Chen, Difei Wang, Wei Wang, Yajie Wang, Michael T. Hemann, Hai Jiang

Corresponding authors: Hai Jiang, Institute of Biochemistry and Cell Biology, Shanghai Institutes for Biological Sciences, Chinese Academy of Sciences and Michael T Hemann, The Koch Institute for Integrative Cancer Research at Massachusetts Institute of Technology

Review timeline:

Submission date:	28 August 2014
Editorial Decision:	30 September 2014
Revision received:	29 December 2014
Editorial Decision:	20 January 2015
Revision received:	06 February 2015
Accepted:	12 February 2015

Transaction Report:

Editor: Roberto Buccione

1st Editorial Decision

30 September 2014

Thank you for the submission of your manuscript to EMBO Molecular Medicine. We have now heard back from the three Reviewers whom we asked to evaluate your manuscript.

You will see that while two Reviewers are clearly supportive of your work one is quite negative. All considered, some issues are raised that prevent us from considering publication at this time. I will not dwell into much detail, as the evaluations are detailed and self-explanatory and will just mention a few main points.

Reviewer 1 would like to see more evidence on the possible side effects of the new compound (providing IHC for instance and treating primary cells in culture) and would also like more details on the mechanism of HDAC inhibition. This Reviewer also lists a number of other items that require your action.

Reviewer 2 is quite positive and has only minor issues for your consideration and action.

Reviewer 3, instead, is quite negative and challenges the notion of the usefulness of an even more potent DNA-damaging agent when the field is currently moving towards finding targeted, non DNA-damaging agents. His/her enthusiasm as a consequence is very low although s/he does admit that your work is "experimentally sound". As for specific issues, Reviewer 3 notes, similarly to Reviewer 1, that more data on HDAC inhibition is required and that data for compounds A and B need to be shown.

Considered all the above, while publication of the paper cannot be considered at this stage, we would be pleased to receive a revised submission, with the understanding that the Reviewers' concerns must be fully addressed with additional experimental data where appropriate and that acceptance of the manuscript will entail a second round of review. As for Reviewer 3's general negative stance on the usefulness of such an inhibitor, this will not be a reason for rejection, but I would encourage you to provide a reply in your rebuttal and also perhaps better discuss the potential usefulness of such a potential drug in the manuscript. Again, to show data on the undesirable side effects on normal proliferating cells will be instrumental to discuss this aspect.

Please note that it is EMBO Molecular Medicine policy to allow a single round of revision only and that, therefore, acceptance or rejection of the manuscript will depend on the completeness of your responses included in the next, final version of the manuscript.

As you know, EMBO Molecular Medicine has a "scooping protection" policy, whereby similar findings that are published by others during review or revision are not a criterion for rejection. However, I do ask you to get in touch with us after three months if you have not completed your revision, to update us on the status. Please also contact us as soon as possible if similar work is published elsewhere.

We look forward to reading your revised manuscript as soon as possible.

***** Reviewer's comments *****

Referee #1 (Remarks):

A DNA/HDAC dual targeting drug with significantly enhanced anticancer potency.

In this manuscript the authors describe their findings concerning the mechanism of action of a bendamustine derivative CY190602 (CY). CY appears to have two functional groups, a nitrogen mustard group driving DNA damage and a side chain that bears homology to hydroxamic acid based HDAC inhibitors. Functional analysis of CY and CY-"mutants" showed that the HDAC-like group inhibits HDAC-activity, induces acetylation of histone H3 lysine residues and acts synergistically with bendamustine in killing cancer cells. Gene transcription analysis of cells treated with CY identified deregulation of genes involved in DNA repair suggesting that CY causes DNA damage and at the same time impairs DNA damage repair, thereby creating a vulnerable situation that is detrimental for cancer cells. Finally the authors show that CY has therapeutic efficacy in a BCR-ABL driven tumor model in mice.

Although I'm in favor of publication of this manuscript, the authors need to address the issues raised below:

Remaining questions:

- Although the authors convincingly show the potency of this compound exceeds bendamustine, a major issue with bringing this compound to the clinic might be that it causes undesirable side effects in normal proliferating cells. The authors could address this issue by treating primary cells (MEFs, ES-cells, HMECs, primary blood cultures, etc) to see how CY effects normal cells. Alternatively, the authors should share some of the histology of mice treated with CY.
- In figure 2 the authors show that CY can inhibit HDAC-activity and induces acetylation of histone H3 lysine residues. These experiments were performed using CY (fig. 2a,b) and do not exclude the possibility that the nitrogen-mustard group contributes to the observed effects. Why did the authors not use CpA and CpB to show that it is the CY linker that inhibits HDAC-activity and induces H3 acetylation (figure 2b).
- In Figure 3 the authors claim that CY results in down regulation of genes critical for DNA damage repair. However several questions remain:
 - o Is the down regulation of the identified genes a consequence of HDAC-inhibition or a consequence of cell death. This might be addressed by performing qPCR of the genes in a time course experiment.
 - o knockdown of the DNA damage repair genes is shown by qPCR but no evidence is provided what

the effect is on protein expression

o Figure 3c shows that knockdown of each of the DNA damage repair genes results in synergy with bendamustine in killing tumor cells. However it is not excluded that knockdown of these genes only results in cell death. The authors should provide this data.

o Do the authors have an explanation why knockdown of all of the tested genes (Tip60, Cbp1, Morf and Msl) do result in synergy with bendamustine?

Referee #2 (Remarks):

This manuscript describes the chimeric HDAC inhibitor connected to a nitrogen mustard that allegedly enhance anticancer potency. The explored compound, CY190602 (CY), is a derivative of bendamustine, the main difference being a 7C membered tail with a hydroxamic acid. These modifications help the compound fit within the HDAC pocket. The biggest improvement is that the drug is seen to act as an HDAC inhibitor while inhibiting DNA repair, serving a dual purpose. The structural importance of both the hydroxamic acid tail and nitrogen mustard moiety were explored and proven to be necessary for the full effect of a dual action drug. The study shows the down-regulation of many DNA repair genes after dosage of CY, as well as the increased potency of CY compared to SAHA and many other HDAC inhibitors. This bendamustine derived compound, CY, shows exciting results for a 'DNA/HDAC dual targeting anticancer drug'. The manuscript is acceptable for publication in EMBO with some minor revisions:

Revisions:

Title - include the compound name within the title

Abstract - line 11: moiety to moieties

Introduction - line 11: DSBs is already abbreviated

Line 18: use FA, since the abbreviation is already given

Line 26: lower case follicular

Results - line 64: "...a gene associated with..."

Line 77: MTD abbreviation not defined "maximum tolerated dose"

Line 95: lower case "...dasatinib..."

Discussion - line 3: "...in vitro; however, the..."

Line 4: "...remained undefined."

Line 14: omit "Because", possible rewrite of the sentence (e.g. Due to both activities being tethered in the same molecule for CY, it is...)

Line 27: "...together, data suggested..."

Line 40: type = types

Line 54: sentence ending in triggered is not a finished thought...please rewrite this sentence.

Line 63: can abbreviate as HDAC

Line 73: of should be in

Line 79: omit indeed

Line 80: omit much

Line 81: omit such

Line 84: omit indeed

Materials and methods

Cell viability assays and determination of relative drug resistance:

Line 6: in = by

Line 14: "...using this previously described method." Jiang et al is already mentioned so does not need to be twice.

Mouse experiments

Line 3: Should define MTD earlier so only an abbreviation is needed here

Line 4: omit of

Figures

All figures have lower case letters associated with the different panels but within the paper they are referred to with upper case letters. Stay consistent with labeling, within figures change to an upper case system

Referee #3 (Remarks):

An article by Liu et al., "A DNA/HDAC dual-targeting drug with significantly enhanced anticancer potency" describes a new chemical compound (CY190602 or CY) which has been derived from bendamustine, a nitrogen mustard type DNA-damaging drug. The CY compound showed enhanced anticancer activity, that is toxicity for cancer cell lines and xenografts. The mechanism behind this increased anti-cancer potency appears to be in blocking of HDACs (histone deacetylases) activity, at least class 1 HDACs, which results in increase in histone acetylation and downregulation of gene expression for a large list of DNA repair and DNA synthesis.

An increase in H3K9ac and H3K56Ac signal clearly suggests problem with both de-acetylation and DNA repair efficiency. The authors propose that the nature of the drug (CY) allows for both DNA damage and HDAC inhibition.

Though the results appear to be experimentally sound I have a few concerns regarding this paper. My prime concern is with the overall concept of creating even more potent DNA-damaging and therefore even more cancerogenic on its own drug whereas majority of cancer research field tries to identify anticancer drugs and treatments that do not invoke DNA damage. For that reason I see no novelty or excitement in proposed by authors potential treatment development by such dual action drug.

I have also questions on why authors selectively show data for one or another compound A or B, or only CY and bendamustine in Figure 1 (panel C, B and D) - what is the DNA damage levels for CY in Fig. 1d experiment? And what are the data for compounds A and B in Figure 1b? Similar questions for Figure 2b and 2d.

Furthermore, though authors demonstrated that they can dock CY onto HDAC active site, one would like to see data on inhibiting HDAC activity in vitro to support their model. Would be also interesting to see if other HDACs e.g. sirtuins that actively participate in modifying DNA repair protein are also inhibited by CY.

1st Revision - authors' response

29 December 2014

A DNA/HDAC dual targeting drug with significantly enhanced anticancer potency.

In this manuscript the authors describe their findings concerning the mechanism of action of a bendamustine derivative CY190602 (CY). CY appears to have two functional groups, a nitrogen mustard group driving DNA damage and a side chain that bears homology to hydroxamic acid based HDAC inhibitors. Functional analysis of CY and CY-"mutants" showed that the HDAC-like group inhibits HDAC-activity, induces acetylation of histone H3 lysine residues and acts synergistically with bendamustine in killing cancer cells. Gene transcription analysis of cells treated with CY identified deregulation of genes involved in DNA repair suggesting that CY causes DNA damage and at the same time impairs DNA damage repair, thereby creating a vulnerable situation that is detrimental for cancer cells. Finally the authors show that CY has therapeutic efficacy in a BCR-ABL driven tumor model in mice.

Although I'm in favor of publication of this manuscript, the authors need to address the issues raised below:

Remaining questions:

- Although the authors convincingly show the potency of this compound exceeds bendamustine, a major issue with bringing this compound to the clinic might be that it causes undesirable side effects in normal proliferating cells. The authors could address this issue by treating primary cells (MEFs, ES-cells, HMECs, primary blood cultures, etc) to see how CY effects normal cells. Alternatively, the authors should share some of the histology of mice treated with CY.

We agree with the reviewer's suggestion and have made efforts to address this question. Our additional experiments showed that:

1) The GI50 of CY is 57uM in MEFs, which is about 18 fold higher than the 3.2uM average GI50 it showed in NCI 60 cancer cell panels (Figure 1B). This suggests that CY preferentially kills cancer cells.

The GI50 of Bendamustine's in MEFs is around 390uM. In the NCI 60 test, for most cancer cell lines 100uM (the cutoff concentration) did not reach GI50 (Figure 1B).

These results were described on page 4, line 9-10.

2) We also tested CY's efficacy in fresh human chronic lymphoblastic leukemia (CLL) cells and healthy B cells (Figure 5b). Although healthy B cells were also affected by CY, at 5uM all CLL cells were killed by CY, whereas B cells were still about 40% viable at 20uM of CY, again supporting a therapeutic window for CY as an anticancer agent.

These results were described on page 8, the second paragraph.

3) In a patient derived xenograft mouse model of human AML, treatment of mice with CY nearly eliminated AML cells, but showed minimal effect on mouse CD45+ (hematopoietic lineage) cells. In contrast, Cytarabine, a first line AML drug, nearly wiped out CD45+ cells but didn't kill AML cells as quickly, or as thoroughly as did CY. This experiment was done by other collaborators and regrettably we don't have it for publication, however it again argues for CY's potential efficacy as a novel anticancer agent.

- In figure 2 the authors show that CY can inhibit HDAC-activity and induces acetylation of histone H3 lysine residues. These experiments were performed using CY (fig. 2a,b) and do not exclude the possibility that the nitrogen-mustard group contributes to the observed effects. Why did the authors not use CpA and CpB to show that it is the CY linker that inhibits HDAC-activity and induces H3 acetylation (figure 2b).

We thank the reviewer for pointing this out. We have performed western blot using all relevant compounds to show that CY and CpA inhibited HDAC, but Bendamustine and CpB did not, which is consistent with our argument. This result was shown in Figure 2b.

- In Figure 3 the authors claim that CY results in down regulation of genes critical for DNA damage repair. However several questions remain:

- Is the down regulation of the identified genes a consequence of HDAC-inhibition or a consequence of cell death. This might be addressed by performing qPCR of the genes in a time course experiment.

As suggested by the reviewer, we performed such experiment, treating cells with various drugs for 6, 8 and 10 hrs. The results showed that there was significant suppression of the identified genes by CY and SAHA at 6 hrs post treatment, yet cell death did not occur up till 10 hrs. This indicated that the down regulation of the identified genes was caused by HDAC-inhibition, but not as a consequence of cell death. These data were included as Fig 3b.

- knockdown of the DNA damage repair genes is shown by qPCR but no evidence is provided what the effect is on protein expression

During the revision, we ordered several antibodies against these genes to analyze knockdown of protein levels, however only CBP antibody showed signal in western blot (Supp. Fig 3), and proved significant knockdown at protein level. We apologize for not being able to provide knockdown data for other proteins. The knockdown vector we used is a microRNA30-based vector, developed by Dr. Greg Hannon and Dr. Scott Lowe, and is very potent for gene suppression. In our previous experience of using this RNAi vector with about 40 other genes, nearly all genes showed significant knockdown at protein level, if corresponding qPCR analysis showed significant knockdown. Based on this we think it is likely that the shRNA we used knocked down target gene at protein levels too.

- Figure 3c shows that knockdown of each of the DNA damage repair genes results in synergy with bendamustine in killing tumor cells. However it is not excluded that knockdown of these genes only results in cell death. The authors should provide this data.

We thank the reviewer for pointing this out. Knockdown of these genes by itself does not affect cell viability, and we have provided this data in Supp. Fig 4.

- Do the authors have an explanation why knockdown of all of the tested genes (Tip60, Cbp1, Morf and Msl) do result in synergy with bendamustine?

The involvements of Tip60, Cbp1 and Msl1 in DNA damage repair have been demonstrated by several groups, and the relevant publications have been referenced in the manuscript. Suppression of these genes will lead to reduced capacity to repair DNA and synergy with bendamustine.

To our knowledge MORF's involvement in DNA repair has not been specifically reported yet. However due to its ability to acetylate Histone and alter histone codes and chromatin structure, it is possible that its deficiency may affect DNA repair. In our experiment result suppression of MORF does confer sensitivity to bendamustine.

Referee #2 (Remarks):

This manuscript describes the chimeric HDAC inhibitor connected to a nitrogen mustard that allegedly enhance anticancer potency. The explored compound, CY190602 (CY), is a derivative of bendamustine, the main difference being a 7C membered tail with a hydroxamic acid. These modifications help the compound fit within the HDAC pocket. The biggest improvement is that the drug is seen to act as an HDAC inhibitor while inhibiting DNA repair, serving a dual purpose. The structural importance of both the hydroxamic acid tail and nitrogen mustard moiety were explored and proven to be necessary for the full effect of a dual action drug. The study shows the down-regulation of many DNA repair genes after dosage of CY, as well as the increased potency of CY compared to SAHA and many other HDAC inhibitors. This bendamustine derived compound, CY, shows exciting results for a 'DNA/HDAC dual targeting anticancer drug'. The manuscript is acceptable for publication in EMBO with some minor revisions:

ions:

Revisions:

Title - include the compound name within the title

Abstract - line 11: moiety to moieties

Introduction - line 11: DSBs is already abbreviated

Line 18: use FA, since the abbreviation is already given

Line 26: lower case follicular

Results - line 64: "...a gene associated with..."

Line 77: MTD abbreviation not defined "maximum tolerated dose"

Line 95: lower case "...dasatinib..."

Discussion - line 3: "...in vitro; however, the..."

Line 4: "...remained undefined."

Line 14: omit "Because", possible rewrite of the sentence (e.g. Due to both activities being tethered in the same molecule for CY, it is...)

Line 27: "...together, data suggested..."

Line 40: type = types

Line 54: sentence ending in triggered is not a finished thought...please rewrite this sentence.

Line 63: can abbreviate as HDAC

Line 73: of should be in

Line 79: omit indeed

Line 80: omit much

Line 81: omit such

Line 84: omit indeed

Materials and methods

Cell viability assays and determination of relative drug resistance:

Line 6: in = by

Line 14: "...using this previously described method." Jiang et al is already mentioned so does not need to be twice.

Mouse experiments

Line 3: Should define MTD earlier so only an abbreviation is needed here

Line 4: omit of

Figures

All figures have lower case letters associated with the different panels but within the paper they are referred to with upper case letters. Stay consistent with labeling, within figures change to an upper case system

We really appreciate the reviewer's positive evaluation of our manuscript and detailed reading of it. We have corrected these parts as suggested.

Referee #3 (Remarks):

An article by Liu et al., "A DNA/HDAC dual-targeting drug with significantly enhanced anticancer potency" describes a new chemical compound (CY190602 or CY) which has been derived from bendamustine, a nitrogen mustard type DNA-damaging drug. The CY compound showed enhanced anticancer activity, that is toxicity for cancer cell lines and xenografts. The mechanism behind this increased anti-cancer potency appears to be in blocking of HDACs (histone deacetylases) activity, at least class I HDACs, which results in increase in histone acetylation and downregulation of gene expression for a large list of DNA repair and DNA synthesis.

An increase in H3K9ac and H3K56Ac signal clearly suggests problem with both de-acetylation and DNA repair efficiency. The authors propose that the nature of the drug (CY) allows for both DNA damage and HDAC inhibition.

Though the results appear to be experimentally sound I have a few concerns regarding this paper. My prime concern is with the overall concept of creating even more potent DNA-damaging and therefore even more cancerogenic on its own drug whereas majority of cancer research field tries to identify anticancer drugs and treatments that do not invoke DNA damage. For that reason I see no novelty or excitement in proposed by authors potential treatment development by such dual action drug.

We thank the reviewer's critical reading of our manuscript. We understand the reviewer's concerns. This issue was also mentioned by reviewer #1. Despite the promising results by targeted therapeutics, many cancer cases do not have targetable oncogenes and currently DNA damaging agents still remains the backdrop of many treatment regimens. For example, In the case of bendamustine, although not a targeted therapeutics, it showed significant clinical benefits in many types of lymphomas and multiple myelomas and gained recent approval for many indications. However in clinics it suffers from low efficacy.

Our effort in generating a more powerful dual-targeting drug may help address some of the issues faced by traditional DNA damaging anticancer drugs. The value of having such anticancer drugs also lies in that, in some case, the existing drug just can't kill cancer cells, which calls for new drug with higher potency. For example, for CLL patients whose tumors carry 17p deletion and lost p53, their lymphoma cells are much more resistant to Bendamustine than lymphoma cells that retain 17p, and clinically Bendamustine fails in these patients. However, as shown below, CY kills 17p-deleted CLL cells (white dots) with similar efficacy to CLL cells that retain 17p (black dots). This suggests that CY may be a potential choice for 17p-deleted CLL treatment. This result was included as Figure 5c and described on page 8, the third paragraph.

With regards to other cancer models, this dual targeting drug also showed promising results. As mentioned in response to reviewer #1's question, in a patient derived xenograft mouse model of human AML, treatment of mice with CY nearly eliminated AML cells, but showed minimal effect on mouse CD45+ cells. In contrast, Cytarabine, a first line AML drug, nearly wiped out CD45+ cells but didn't kill AML cells as quickly, or as thoroughly as did CY. This result again argues for CY's ability to preferentially kill transformed cells and supports its potential efficacy as an novel anticancer agent.

Moreover, with the development of tumor specific drug delivery technologies, a more powerful version of anticancer drug at least offers more powerful choices for antibody-coupled drug delivery technologies, if the drug's ability to preferentially kill cancer cells still remains an issue.

In summary, we hope to convey the idea that dual-targeting cancer drugs like CY may provide prototypes and valuable new tools for additional clinical research and drug development.

I have also questions on why authors selectively show data for one or another compound A or B, or only CY and bendamustine in Figure 1 (panel C, B and D) - what is the DNA damage levels for CY in Fig. 1d experiment? And what are the data for compounds A and B in Figure 1b? Similar questions for Figure 2b and 2d.

We thank the reviewer for pointing this out. We initially only presented the data that we thought are most relevant, however data on other drugs/drug combinations should have been presented as additional positive/negative controls. We have provided western blot data on all relevant compounds and compound combinations in these experiments (Figure 1d). Figure 2d meant to show CpdB's ability to cause DNA damage. Other drugs, such as Bendamustine, CpdA's ability or inability to cause DNA damage, been shown in Figure 1d. For Figure 2e, the experiment analyzed IC50 for individual drugs, and the assay was not suitable for drug combinations.

Furthermore, though authors demonstrated that they can dock CY onto HDAC active site, one would like to see data on inhibiting HDAC activity in vitro to support their model. Would be also interesting to see if other HDACs e.g. sirtuins that actively participate in modifying DNA repair protein are also inhibited by CY.

The in vitro inhibition data with various HDACs was provided in Figure 2A. We looked at published data and it was reported that HDAC inhibitors do not interfere with sirturin activity (Drummond DC, Noble CO, Kirpotin DB, Guo Z, Scott GK, et al. (2005) Clinical development of histone deacetylase inhibitors as anticancer agents. *Annu Rev Pharmacol Toxicol* 45: 495–528.), probably due to the fact that sirturins belong to a different subclass of HDACs. We also looked at the possibility that HDAC inhibition may also downregulate sirtuins, which we did not initially thought of. However upon querying the 12 HDAC-inhibitor microarray datasets within the connectivity map database, none of the sirturin genes were significantly downregulated in any of the 12 experiments.

Thank you for the submission of your revised manuscript to EMBO Molecular Medicine. We have now received the enclosed reports from the referees that were asked to re-assess it. As you will see the reviewers are now globally supportive and I am pleased to inform you that we will be able to accept your manuscript pending some final editorial amendments.

Please submit your revised manuscript within two weeks. I look forward to seeing a revised form of your manuscript as soon as possible.

***** Reviewer's comments *****

Referee #1 (Comments on Novelty/Model System):

I thank the authors for addressing the issues I raised and support the publication of this manuscript in EMBO Molecular Medicine

Referee #3 (Remarks):

the revised manuscript is now of satisfactory quality and suitable for publication.